# Global Climate Change and Indigenous Peoples in Taiwan: A Critical Bibliometric Analysis and Review

Mucahid Mustafa Bayrak [1,2,*] , Yi-Ya Hsu [1] , Li-San Hung [1], Huei-Min Tsai [2,3] and tibusungu 'e vayayana [1,2,4]

1 Department of Geography, National Taiwan Normal University, Taipei City 10610, Taiwan; keziahsu@gmail.com (Y.-Y.H.); lshung@ntnu.edu.tw (L.-S.H.); t24019@ntnu.edu.tw (t.'ev.)
2 Center for Indigenous Research and Development, National Taiwan Normal University, Taipei City 10610, Taiwan; hmtsai@ntnu.edu.tw
3 Graduate Institute of Environmental Education, National Taiwan Normal University, Taipei City 11677, Taiwan
4 Former Deputy Minister, Council of Indigenous Peoples, Executive Yuan, New Taipei City 24220, Taiwan
* Correspondence: mmbayrak@ntnu.edu.tw; Tel.: +886-02-7749-1680

**Abstract:** In recent years, the subject of Indigenous peoples and global climate change adaptation has become a rapidly growing area of international study. Despite this trend, Taiwan, home to many Indigenous communities, has received relatively little attention. To date, no comprehensive review of the literature on Taiwan's Indigenous peoples and global climate change has been conducted. Therefore, this article presents a bibliometric analysis and literature review of both domestic and international studies on Taiwan's Indigenous peoples in relation to resilience, climate change, and climate shocks in the 10-year period after Typhoon Morakot (2009). We identified 111 domestic and international peer-reviewed articles and analyzed their presentation of the current state of knowledge, geographical and temporal characteristics, and Indigenous representation. Most studies were discovered to focus on post-disaster recovery, particularly within the context of Typhoon Morakot, as well as Indigenous cultures, ecological wisdom, and community development. This study also discovered relatively few studies investigating how traditional ecological knowledge systems can be integrated into climate change adaptation. Most studies also adopted a somewhat narrow focus on Indigenous resilience. Large-scale quantitative and longitudinal studies are found to be in their infancy. We observed a geographical skewness among the studies in favor of southern Taiwan and relatively limited engagement with contemporary studies on Indigenous peoples and climate change. We furthermore determined a large overlap between the destruction path of Morakot and study sites in the articles. Indigenous scholars have managed to find a voice among domestic and international outlets, and an increasing number of scholars have argued for more culturally sensitive approaches to post-disaster recovery and disaster management in Taiwan.

**Keywords:** Taiwan; indigenous peoples; resilience; global climate change; bibliometric analysis; Typhoon Morakot

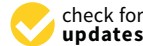



## 1. Introduction

The world's Indigenous peoples represent less than 5% of the global population, yet they currently manage or have rights over numerous ecosystems, ranging from the Arctic to the tropical rain forests of Borneo [1]. As the current and projected consequences of global climate change and environmental change come under greater scrutiny, academics have noted that the world's Indigenous peoples bear little responsibility (if any) for the forecast consequences [2]. This is the reason for the rapidly growing trend in studies on Indigenous peoples, resilience, and global climate change [1,3–6]. Such studies have explored how Indigenous peoples' knowledge systems, institutions, worldviews, conservation practices, and local perceptions could be of benefit or be integrated into climate change adaptation

and mitigation programs. These studies have had many aims, including exploring Indigenous alternatives for sustainable ecosystem management [7], understanding Indigenous peoples' perceptions of climate change [4], and Indigenous peoples finding representation in global climate change debates and negotiations [8]. Research has been conducted on various levels. Microlevel studies have often focused on particular Indigenous communities, traditional ecological knowledge (TEK) systems, climate resilience and livelihood adaptations, and local perceptions of climate change [9–12]. Mesolevel studies tend to investigate Indigenous conservation practices and land management on a regional or cross-country level [13–16]. Lastly, macrolevel studies have either performed a global analysis of previous studies [4,6,17] or adopted a global approach to Indigenous peoples' research and global climate or environmental change [1,8].

Within the context of the growing trend in research on Indigenous peoples and global climate change, we now consider Taiwan, which is home to 16 officially recognized Indigenous groups as well as other locally or unofficially recognized groups (Figure 1; Appendix A). Taiwan's Indigenous peoples (Táiwān yuán zhù mínzú) accounted for 573,086 people in 2020 (2.4% of the island's total population), of whom 287,789 lived in an Indigenous community [18]. Taiwan's Indigenous peoples are Austronesian, and some communities have been able to conserve their culture, customs, traditional livelihoods, and practices despite centuries of colonialization, assimilation, and suppression [19–21].

Global climate change poses a considerable challenge for Taiwan. Since the 1990s, there has been a growing awareness of the impacts of climate change on the nation. This started with the devastating effects of Typhoon Herb in 1996 (73 fatal, 463 non-fatal causalities) and Typhoon Nari in 2001(104 fatal, 265 non-fatal causalities). Climate change is expected to increase temperatures and heatwave frequency throughout the country. Rainy seasons will bring more precipitation, whereas dry seasons will become drier, and typhoons and associated extreme rainfall events are expected to increase in intensity, although not necessarily in frequency [22,23]. Due to its location in the Asia-Pacific, Taiwan regularly experiences climate events that have a negative impact. Of the 384 recorded instances of extreme climate events that had a negative impact on Taiwan between 2006 and 2020, 43.2% occurred or directly impacted Indigenous communities [24]. Taiwan's Indigenous peoples are therefore disproportionately exposed to the negative effects of climate events [22,25,26]. Typhoon Morakot, which struck central and southern Taiwan in August 2009, is perhaps most exemplary of the destructive effects of climate change on Indigenous and rural communities to date [27]. The typhoon killed 699 people, destroyed 1766 houses, and displaced 4500 residents [28,29]. After this national tragedy, numerous studies were undertaken to investigate the effect of climate change on Taiwan's Indigenous peoples, and relevant articles have been published through both domestic and international publishing outlets.

An international conference entitled "Climate Change, Indigenous Resilience, and Local Knowledge Systems: Cross-Time and Cross-Boundary Perspectives" was organized by the Research Institute for the Humanities and Social Sciences and took place in December 2019 in Taipei City. This conference explored the state of knowledge on climate change and Indigenous resilience in Taiwan 10 years after Typhoon Morakot. One critical issue raised during this conference was the underrepresentation of studies from Taiwan in the international literature. Whether this was due to a lack of Taiwanese studies or Taiwan being largely overlooked by international scholars is unclear. Therefore, the primary aim of this study was to assess the state of knowledge of Indigenous peoples and climate change in Taiwan since Typhoon Morakot. This was achieved through bibliometric analysis and a literature review of articles published in both domestic and international peer-reviewed academic journals and books over the past 10 years.

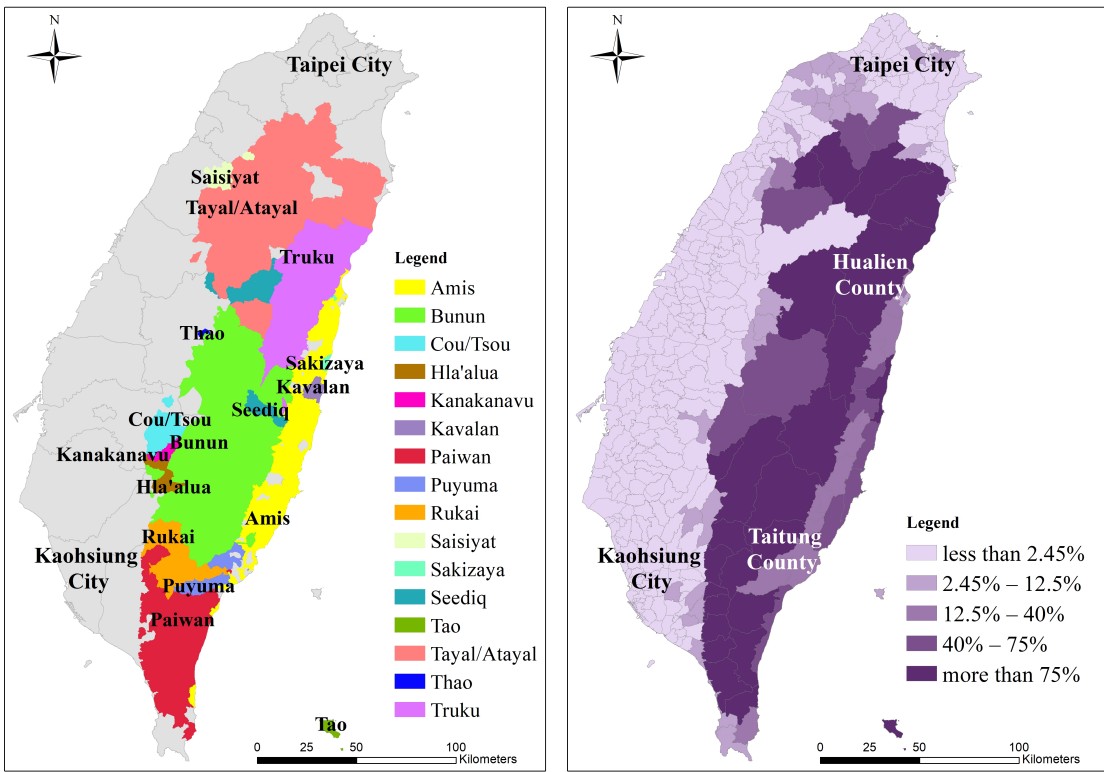

**Figure 1.** (**a**) Distribution of recognized Indigenous areas in Taiwan (data from: [18]) and (**b**) the percentage of Indigenous population in Taiwan on district/township level (data from: [30]).

## 2. Research Questions and Methods

The aims of this study could be subdivided into five research questions: (1) what trends, themes, and topics can be found among domestic and international studies, and what differences can be observed; (2) what is the geographical distribution of the studies in Taiwan, and which Indigenous groups have been selected; (3) what are the temporal characteristics of the studies (ranging from risk perception to post-disaster recovery); (4) to what extent are Indigenous Taiwanese voices represented in such academic articles; (5) what knowledge gaps and potential research directions can be identified.

We performed bibliometric analysis (employing similar methods as: [6,31,32]) using the databases of Scopus and Airiti Library. Several Taiwanese journals are not indexed in Scopus, which was why Airiti Library was selected for finding Chinese-language articles. The time frame of our research was from January 2010 until April 2020. This date was chosen as we started a special issue on the same theme [33], and this would thus heavily influence our bibliometric analysis. We used Boolean search strings, shown in Table 1, to identify articles that referred to climate change, Indigenous peoples, disaster, or resilience in their titles, abstracts, or keywords. Climate change could also refer to climate hazards or disasters, climatic change, or climate variability. When we searched for international articles, we also employed Taiwan as a keyword to geographically restrict our search results. Additionally, we employed a snowball method in our literature review, examining the references of all identified studies to find other relevant studies. In addition to the inclusion criteria above, this project concerned itself with only peer-reviewed studies.

**Table 1.** Search terms for the bibliometric analysis.

| Audience & Database | Search Terms (Climate Change and Indigenous Resilience Focus) | | | | |
|---|---|---|---|---|---|
| International Scopus | "Taiwan" | AND | "Indigenous" or "Indigenous people" or "aboriginal" | AND | "disaster" or "climate" or "hazard" or "local knowledge" or "resilience" |
| Domestic Airiti Library | | | "Indigenous people" (原住民, Yuán zhùmín) or "tribe" (部落, Bùluò) | AND | "disaster" (災害, zāihài) or "disaster" (災難, zāinàn) or "climate change" (氣候變遷, qìhòu biànqiān) or "resilience" (韌性, rènxìng) or "disaster" (災, zāi) |
| Inclusion criteria | 1. climate change and Indigenous resilience focus<br>2. 2010 January -2020 April<br>3. Keywords AND Title AND Abstract | | | | |

All studies that did not focus (this could range from being the main focus to being relevant to the topics) on climate change (or climate disasters), resilience, or Indigenous peoples were excluded from our constructed database. It is important to take into account that Indigenous peoples in Taiwan also face other negative (environmental) events such as earthquakes, land subsidence, or tsunamis. Therefore, this study does not consider all disasters. At the same time, it is important to note that Indigenous resilience could be applied to all stressors and shocks [34]. Furthermore, climate events and disasters could be considered to be climate change-related but cannot always be proven to be caused by global climate change.

After identifying the relevant articles for the literature review, a dataset in Microsoft Excel was created in which the articles were categorized on the basis of the year of publication, language, type of disaster, the ethnicity of the studied group, the ethnicity of the authors, themes of the study, phase in disaster management, and geographical distribution among other items. Data were analyzed using Microsoft Excel and Power BI [35]. The relevance of the data was assured by conducting a literature review to better understand the trends and themes among the articles in our database, and geographic information system techniques combined with secondary data were employed to identify the geographical distribution and knowledge gaps of the studies. Power BI was employed to visualize and analyze the papers, themes, and topics, and an interface was designed enabling the user to interact with and analyze the data themselves (Appendix B).

## 3. Results and Discussion

In total, we discovered 111 articles, 50 of which were indexed in Scopus (labeled as international articles) and 61 in Airiti Library (domestic articles; Figure 2; Appendix C). Each year saw an increase in the number of articles, and the number of domestic and international articles peaked in 2012 and 2016, respectively. The international articles were published in *Sustainability* (Switzerland; n = 4), *Natural Hazards* (n = 2), and other journals ranging from *Land Use Policy* (n = 1) to *International Psychogeriatrics* (n = 1). Most of the journals were either related to disaster management (e.g., the *International Journal of Disaster Risk Reduction*) or sustainability sciences (e.g., *Sustainability Science*). The domestic studies were published in the *Journal of Slopeland Hazard Prevention* (n = 6), *Taiwan: A Radical Quarterly in Social Studies* (n = 5), the *Journal of Natural and Human Environment of Indigenous Peoples* (n = 4), and the *Journal of the Taiwan Indigenous Studies Association* (n = 4). Other domestic articles were published in various social and natural science journals. At least four of the Taiwanese journals specifically focus on Indigenous peoples; in addition to those aforementioned, these included the *Taiwan Indigenous Studies Review* and *Taiwan Journal of Indigenous Studies*.

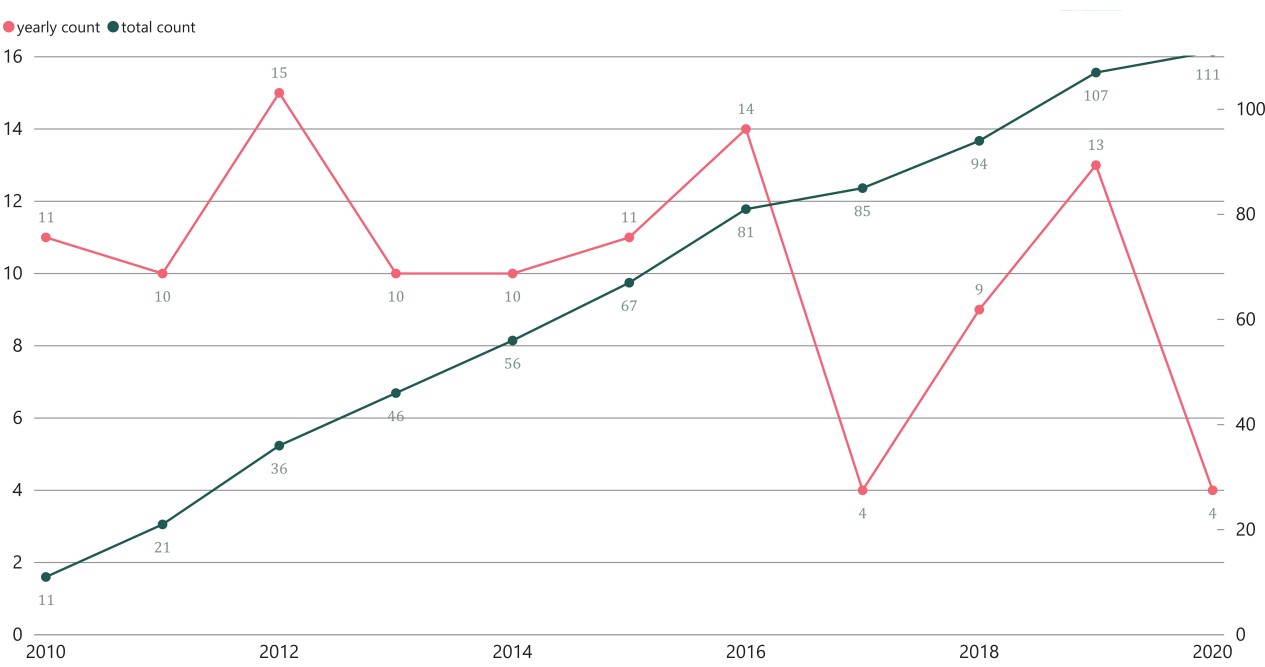

**Figure 2.** Number of articles by year (frequency, left; cumulative count, right).

### 3.1. Trends, Themes, and Topics

The studies in our database were categorized into at least one of the following 10 themes and topics: disaster management; Indigenous culture (including cultural practices, traditions, institutions, and worldviews); ecological wisdom; community development; housing and sustainable architecture; Indigenous health; Indigenous tourism; sustainable agriculture; climate justice; and education (Figure 3).

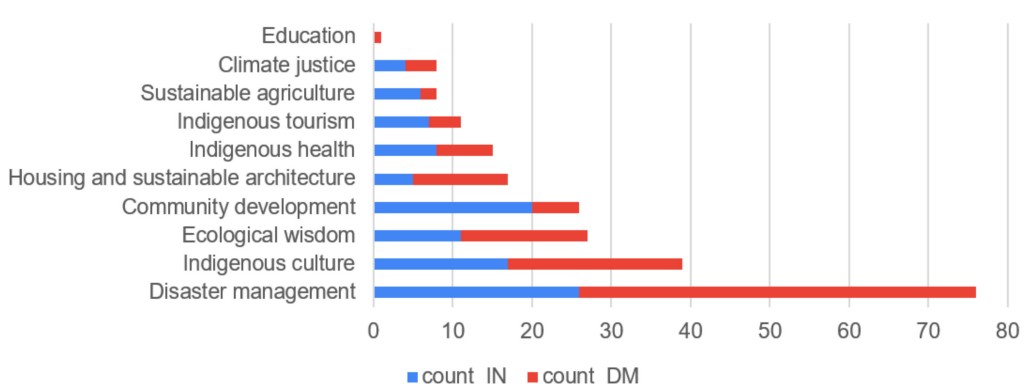

**Figure 3.** Themes and topics of the articles (frequency; multiple options possible). Note: count_IN = international articles, count_DM = domestic articles.

After Typhoon Morakot, there was a surge in studies on the impact of the typhoon on Indigenous peoples as well as the post-disaster recovery efforts (Table 2), especially in the years 2012 and 2016, as then most projects would come to an end and results were consequently presented. These studies focused on issues ranging from lessons learned from relocation and resettlement policies [36] to cultural issues in post-disaster reconstruction [28]. Cultural issues (the second most common theme) were considered to be crucial in both domestic and international studies. Some of these articles are critical of the Taiwanese government's response after Typhoon Morakot. After Morakot, many Indigenous communities in southern Taiwan were relocated to or resettled in new locations. Many scholars

argued that these government policies were insensitive toward Indigenous cultures and historical vulnerabilities [29,37–40]. Various Indigenous groups were relocated together; pre-existing villages to which Indigenous groups were resettled were not accustomed to Indigenous cultures; numerous households were ineligible for governmental housing; and resettled families were unable to continue their farming activities or sell their newly acquired homes [37,41–43].

**Table 2.** Type of climate disaster investigated in the articles (frequency; multiple options possible; count_IN = international articles, count_DM = domestic articles).

| Type of Disaster | Count_IN | Count_DM | Total |
| --- | --- | --- | --- |
| Typhoon Morakot | 25 | 31 | 56 |
| Debris flow/landslide/rockfall (rainfall induced) | 2 | 13 | 15 |
| Typhoon | 1 | 8 | 9 |
| Heavy rain | 0 | 3 | 3 |
| Land subsidence | 0 | 1 | 1 |
| Wind | 0 | 1 | 1 |
| General | 23 | 21 | 44 |

Other studies focused on climatic stressors and shocks: other typhoons besides Morakot; debris flow, landslides, or rock falls as a result of heavy rain; and more general aspects of climate change such as droughts, flooding, and climate variability (Table 2). Some studies created climate resilience or vulnerability indices for urban [44] or rural [25] settings. Several studies focused on the ecological wisdom of Indigenous peoples, including TEK systems [45–47], agroforestry and conservation practices [48–50], the roles of traditional institutions in conservation [51], traditional housing and settlement patterns [52], and traditional knowledge and risk perception [53]. Even though many studies acknowledged the importance of TEK systems, only a few focused on how TEK could be integrated into climate change adaptation [54–56]. The majority of the studies were published in domestic journals. Notably, Kuan [54] presented a detailed case study on the TEK systems of the Atayal/Tayal people and contemporary disaster management in a watershed area. Wang [56] specifically focused on the perceptions of climate change of the Tayal people and how their TEK systems could support households to identify climate change adaptation options. Lin et al. [52] investigated how Indigenous Tao (or Yami) people employed their ecological wisdom by choosing the appropriate settlement location and housing architecture for coping with strong winds on Lanyu (Orchid) Island.

Other major topics and themes included community development (26 articles), housing and sustainable architecture (17 articles), and Indigenous health (15 articles). In terms of community development, some studies referred to either cultural and social vulnerability [28] or procedural vulnerability [38]. The procedural vulnerability concerns the relationships people have with power rather than with the environment [38]. Relocation after a climate disaster, for example, has often been labeled as a double disaster in Taiwan because it shifts Indigenous peoples from one vulnerable situation into another that may be worse [39]. In the health sciences, scholars focused on mental health, posttraumatic stress disorder, and depression among Indigenous peoples after Typhoon Morakot and other climate disasters [57,58]. Chen et al. [59], for example, reported that Indigenous peoples tended to show stronger mental recovery from Typhoon Morakot than Han people (the ethnic majority in Taiwan) due to their higher adaptability to cope with a changing environment and climate. Findings from health science studies thus indicate that research on mental resilience could complement studies on Indigenous peoples' resilience when faced with climate change [60,61].

Indigenous tourism (11 articles) is currently a trending topic among the literature on Indigenous peoples and climate change. A growing amount of research is focusing on how Indigenous tourism contributes to Indigenous resilience against climate disasters [62,63] or contribute to community development [51,64]. Scholars often perceived a relationship

between community or household participation in Indigenous tourism (in its various forms) and enhanced resilience to climate change. However, whether Indigenous tourism can be advantageous to all of the environment, economy, and Indigenous peoples in Taiwan within the context of a changing climate remains unclear.

We created two word clouds—for domestic and international articles, respectively—which are presented in Figure 4. This study identified that the following terms and keywords appeared in international articles (Figure 4a): Disaster, Indigenous, community, Morakot, resilience, vulnerability, and sustainable. The terms resilience and vulnerability did not appear in the keywords of articles published in domestic journals (Figure 4b); such articles had keywords mainly focused on disaster, Morakot, Indigenous, typhoon, and community. Understandably, most international and domestic studies focused on the impact of Typhoon Morakot on Taiwan's Indigenous peoples and were conducted on the microlevel. This study also discovered limited engagement with the recent international literature on Indigenous peoples and climate change. Studies focusing on Indigenous resilience adopted a somewhat narrow understanding by focusing primarily on coping or adaptation strategies, but transformative responses to climate change remain understudied. We could also not find any cross-country analyses (i.e., comparing Taiwan with another country), except for one study comparing Australia with Taiwan [65]. General studies on how Taiwan's Indigenous peoples have experienced or perceived global climate change are also lacking. Although some studies focused on multiple communities within southern Taiwan, no studies could be found that were conducted on a national level. Lastly, a qualitative approach was employed in the majority of studies, sometimes interviewing only a few individuals; large-scale quantitative studies are still in their infancy. The latter is not due to lack of data, as many government-led quantitative studies have been conducted on the impacts of Typhoon Morakot [66] and other major natural disasters.

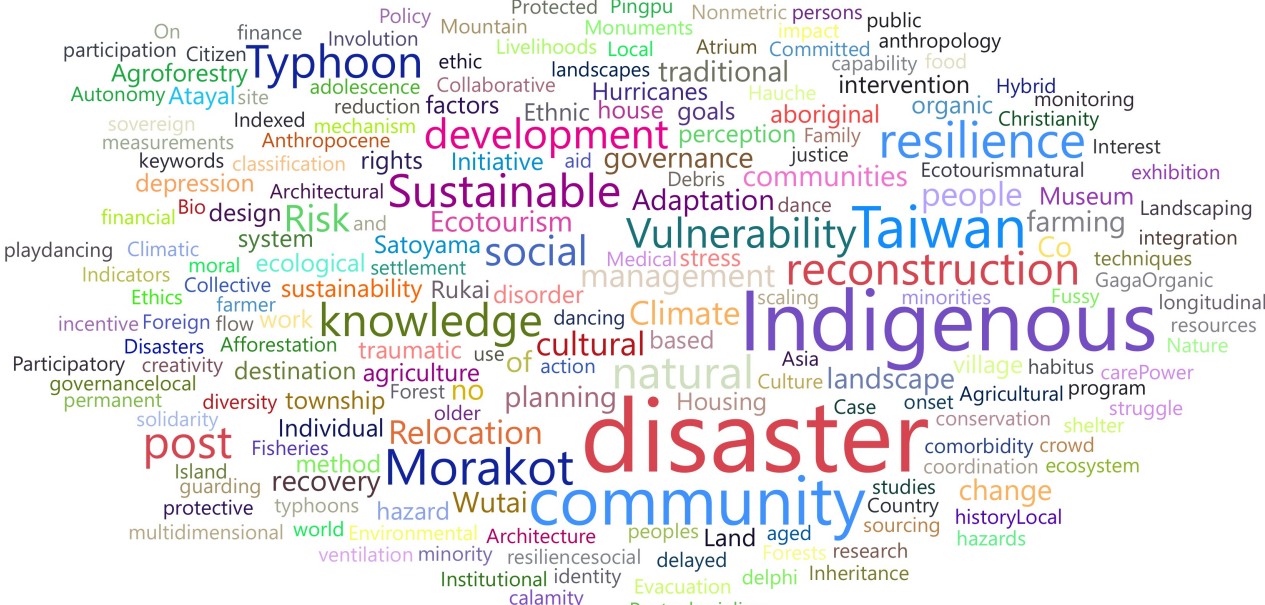

**Figure 4.** *Cont.*

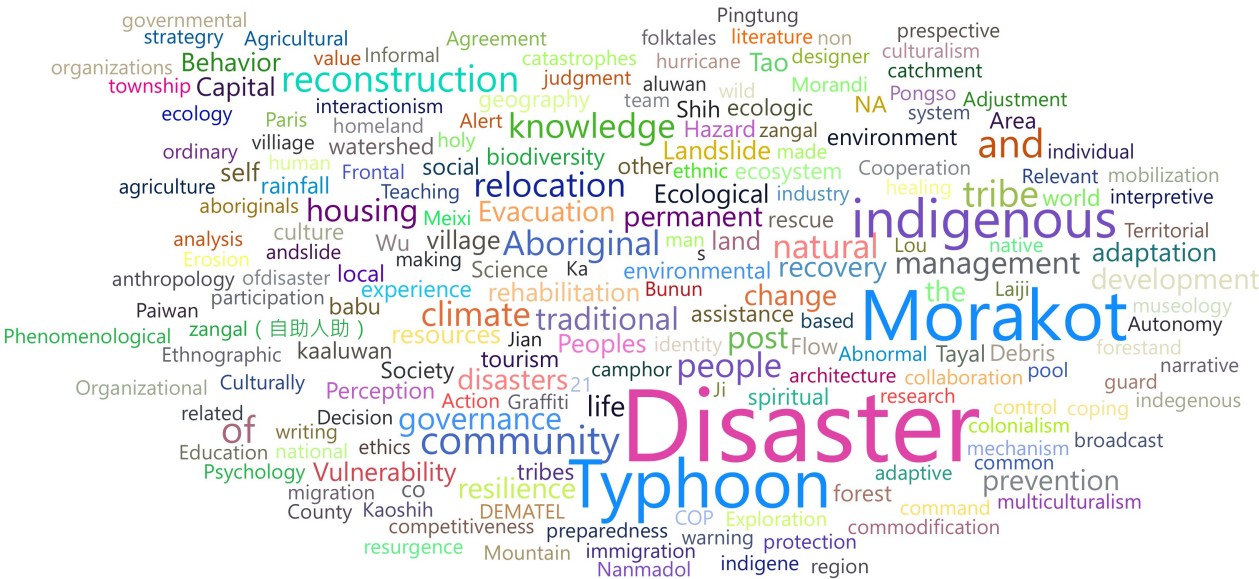

**Figure 4.** Word clouds of international (**a**) and domestic (**b**) articles.

### 3.2. Geographical Distribution and Indigenous Groups

The related studies have been geographically uneven in their focus; some (Indigenous) districts or townships have received considerably more attention than others (Figure 5). This could be related to the distribution of Indigenous peoples in Taiwan, but very few studies have been conducted about the eastern coast of Taiwan (compare Figure 5 with Figure 1). Additionally, domestic (Figure 5a) and international (Figure 5b) studies had different geographical distribution patterns, with international studies showing a higher degree of geographic concentration than their domestic counterparts. Most studies were conducted in southern Taiwan, including Ping Tung and Indigenous areas of Kaohsiung. Many places with a relatively high percentage of Indigenous peoples received little to no attention, whereas some places with a relatively low percentage received much more attention, both domestically and internationally. One notable example—which has been excluded from the database—was a study on local and Indigenous knowledge among coastal Han villages in Buidai township, Chiayi county [67]. The study failed to mention that no Indigenous peoples resided in their selected study sites, but they still investigated the local and "Indigenous" knowledge systems of local households.

Analysis of the geographical distribution of the negative impacts of climate disasters in Taiwan could explain the geographical skewness of the studies in favor of southern Taiwan (Figure 6 and Appendix D). Figure 6a shows a large overlap between the destruction path of Morakot and the sites investigated in both domestic and international studies. As revealed by Figures 5 and 6, Namasia district, a mountainous Indigenous district in the northeastern part of Kaohsiung, was the subject of numerous studies and also among the regions most heavily affected by Morakot. Similar overlaps occurred in sites in northern Ping Tung and other mountainous areas of Kaohsiung. However, when considering the total damage caused by climate disasters and relevant studies (Figure 6b), we observed a somewhat different picture. Many places that have been affected by climate hazards were not represented in any of the studies, which could be related to relatively few causalities. Very few studies [68,69] were conducted in Yilan, Hualien, or Nantou counties in the period of 2010–2020. Another observation is that many studies focused on climate shocks (such as typhoons); less attention was paid to gradual changes or climate stressors. Little information is available in the literature on the climate change resilience and consideration of Indigenous communities on Taiwan's east coast, which is home to many such communities. Figure 6 indicates an overlap between climate shock events and the literature, rather than holistic climate resilience studies being conducted into Indigenous

groups and reflecting gradual changes and stressors in general. A notable exception has been Lanyu/Orchid Island, home to the Tao people. Relevant studies [46,52] specifically focused on this island due to the assumed resilience of the Tao people to negative climate events.

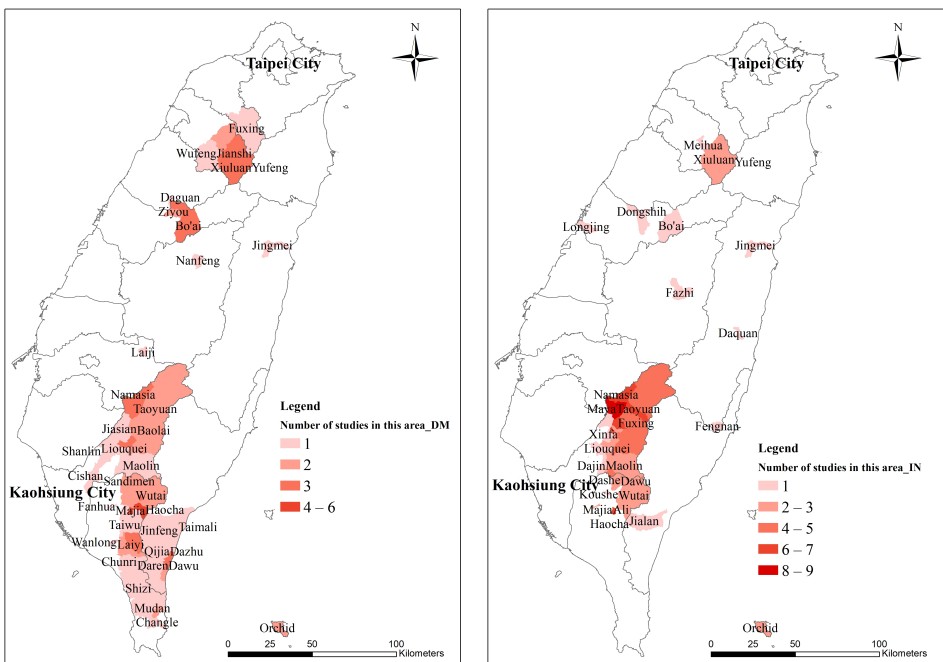

**Figure 5.** Geographical distribution of (**a**) domestic and (**b**) international articles at village/town/district level. *Note:* if studies were conducted in multiple research sites, all sites are shown on the map. Studies that did not refer to a specific research site were excluded.

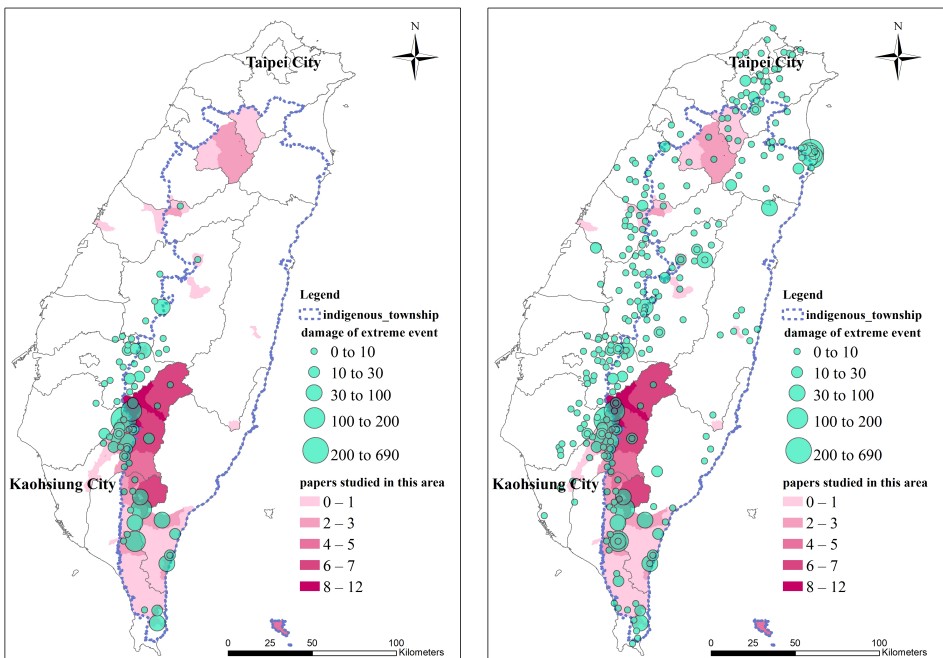

**Figure 6.** Geographical distribution of (**a**) all articles and damage caused by Typhoon Morakot in 2009 and (**b**) all articles and damage caused by extreme climate events on the national level from 2006 till 2020 (data from: [24]). *Note:* damage refers to a cumulative score of deceased and injured people and severely damaged or destroyed houses as a result of extreme climate events, such as typhoons, rainfall-induced debris flows, or heavy rainfall.

Another reason for the uneven geographical distribution among the included studies is the relationship between university locations and site access. Some universities in southern Taiwan (National Ping Tung University, National Sun-Yat Sen University, National Cheng Kung University, and I-Shou University in particular) have extremely active research centers or colleges dedicated to Indigenous studies. Researchers from these universities have produced numerous studies, with articles published both domestically and internationally. This was partly the result of government policy to allocate substantive research funding to southern universities in Taiwan to study the impacts of Typhoon Morakot instead of allocating it to their northern counterparts. In Taipei City, researchers from National Chengchi University and National Taiwan University have taken the lead in conducting studies on Indigenous peoples and those living in north-central Taiwan in particular. According to our database, Wulai, an Indigenous Tayal district in New Taipei City, interestingly received very little attention from Taipei-based scholars over the past 10 years. This is remarkable because Wulai was severely affected by Typhoon Soudelor in 2015 [70], and the district is close to Taipei City. Wulai district has been relatively well prepared for typhoons [70], which partly explains why it has not been a focus in the analyzed articles.

We also searched for the individual Indigenous groups in the relevant studies and identified 13 groups in total (Figure 7): 12 officially recognized groups and one locally recognized group (Taivoan). The Rukai and Paiwan peoples were the most studied with 29 (26.1%) and 28 (25.2%) articles, respectively. These Indigenous groups were most heavily affected by Morakot and government relocation policies [29,39], so it should therefore not come as a surprise that they were the most investigated. The third and fourth most studied groups were the Tayal (17 articles; 15.3%) and the Bunun (14 articles; 12.6%) respectively. Many articles on the Tayal focused on either their agricultural or hunting practice or their TEK systems [45,54,71]. The east coast's Amis people—Taiwan's largest Indigenous group, which accounts for 213,958 people [72]—are underrepresented with only three articles. The Saisiyat, Thao, Sakizaya, and Kavalan peoples were not mentioned in any of the articles, and 21.8% of all articles failed to mention a specific Indigenous group. This is most likely because the authors assumed their readers would know which groups were involved in their study based on the location of the study site or because the authors did not consider this to be relevant information. Some authors also had problems understanding the differences between the terms Indigenous, aboriginal, tribe, and ethnic minority. In one article [58], the aforementioned words were used interchangeably; it was stated that (t)he Indigenous people are the ethnic minority group [sic] in Taiwan (p.12). Hsu [37], in her insightful study on Taiwan's imagined geographies and identities, discusses in great detail the political implications and issues related to the classification and recognition of Taiwan's Indigenous peoples. This is reflected in how some articles (often from outside the social sciences) identify or acknowledge the relevant Indigenous groups. Additionally, the process of translating Chinese terms into the English language could have caused some confusion among scholars [37].

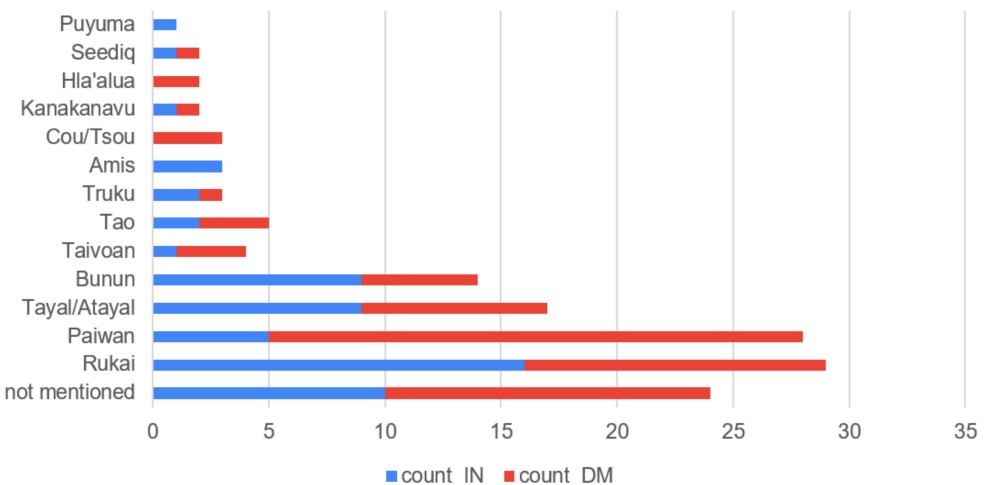

**Figure 7.** Indigenous peoples mentioned in the research articles (frequency; multiple options possible).

### 3.3. Temporal Characteristics

For disaster management literature analysis, we divided all articles into four categories reflecting the temporal characteristics of the disaster phase being investigated: risk perception, disaster risk reduction (DRR), in-season coping strategies, and post-disaster recovery (Figure 8). These categories refer to the temporal orientations of the studies and correspond to the four stages of disaster management (mitigation, preparedness, response, and recovery, respectively). Each phase reflects how climate change affects Indigenous livelihoods and adaptation strategies. Post-disaster recovery was the focus of most articles (76 in total; 68.5%), whereas DRR was the second most reported, risk perception third, and in-season coping strategies last. With the exception of one article [59], the articles in our database did not focus on all four phases of disaster management.

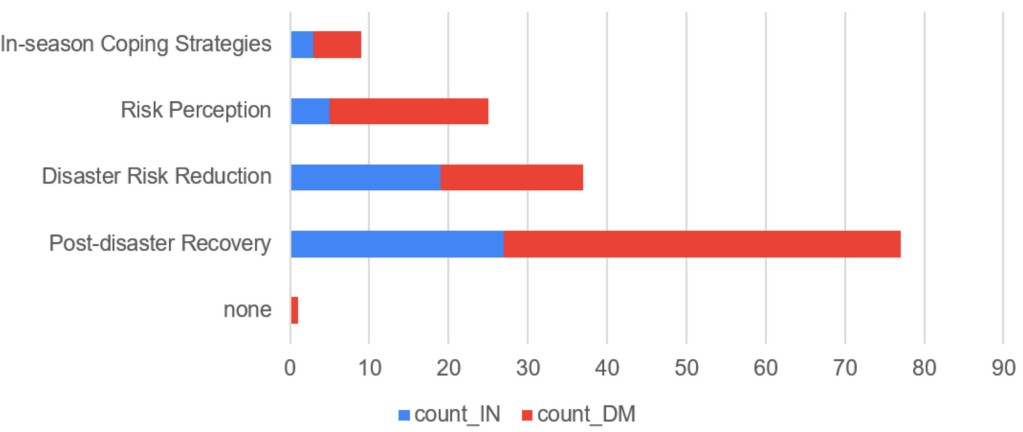

**Figure 8.** Phase of disaster management (frequency; multiple options possible).

In the previous section, we mention that the majority of studies focused on post-disaster recovery or the post-disaster setting. Many articles shared the lessons learned from Taiwan after Typhoon Morakot in a variety of contexts, ranging from the highly descriptive or technical [36] to the very critical [65]. Generally, Taiwan's approach to disaster management is somewhat top–down [39]. This is why numerous scholars have argued for more culturally appropriate post-disaster recovery strategies within Indigenous contexts [28,42,43]. Few studies adopted a longitudinal approach [73], and follow-up studies were rare. Many studies also lacked a clear baseline, but this is somewhat understandable given the unpredictable nature of climate hazards.

Although many of the studies could be considered to have investigated the DRR or risk perception phases, few studies elucidated how TEK could play a role in both perceiving and preparing for climate disasters and climate change more generally [48,53,54,71]. Roder et al. [53] included a relatively small section on traditional knowledge and risk perception in a Bunun community in their article, whereas Ba et al. [48] investigated the Rukai's traditional farming methods for coping with disasters and achieving sustainable development. Another insightful study, conducted in an Indigenous community in the mountains of Taichung county, revealed that local Tayal households could identify disaster risks by detecting changes in local terrain, hydrology, flora, and fauna [71]. However, the lack of relevant studies indicates that more studies on TEK in relation to disaster management should be conducted.

Only nine studies (8.1%) shed light on in-season coping strategies. For instance, a study conducted in (Indigenous) mountainous villages in Kaohsiung City revealed that 86.2% of the households did not receive any formal early disaster warning of the onset of debris flows during Typhoon Morakot [74]. These households solely relied on their intrinsic senses and Indigenous knowledge before and during the disaster. An article published in the *Fooyin Journal of Health Sciences* revealed the power struggles between Indigenous peoples and governmental medical personnel during Typhoon Morakot and its direct aftermath. These struggles were caused by mainstream societal misperceptions of the inferiority and vulnerability of Indigenous peoples in Taiwan [75]. However, Indigenous experiences and strategies during climate shocks and stressors generally remain underinvestigated.

### 3.4. Indigenous Authors and Voice

Various scholars increasingly agree that Indigenous voices deserve a more prominent role in the Conference of the Parties (COP) meetings, Intergovernmental Panel on Climate Change (IPCC) reports, and academic studies [8,31]. Taiwan, which is internationally unrecognized by the vast majority of the world's nations, is perhaps not active in COP meetings, but numerous academic studies have been written by Indigenous scholars. At least 22.9% (14 studies) of domestic articles were either written or co-written by Indigenous authors. For international articles, this percentage was at least 12.0% (six studies in total). We identified Indigenous scholars by their names or their ethnicity being otherwise mentioned in the main text. These Indigenous scholars self-identified as Rukai, Paiwan, Puyuma, Tayal, Seediq, Tao, Sakizaya, Amis, and Tsou/Cou. They either wrote the articles as a single author or together with other Taiwanese, Indigenous, or foreign scholars. In terms of diversity among the Indigenous authors, nine groups are represented, meaning that seven groups (e.g., the Bunun) are not represented. In total, seven non-Taiwanese scholars contributed to studies on Taiwan's Indigenous peoples and climate change; two of these studies were co-written by Indigenous scholars. Given that Indigenous peoples only make up 2.4% of Taiwan's total population, Taiwan's Indigenous scholars can be concluded to have found a voice in academia, especially in articles published in domestic outlets (Appendix C). It is hoped that more domestic research can be translated into English for an international audience because many articles written by Indigenous scholars contain very detailed ethnographies on local TEK systems [47,54,55].

### 3.5. Future Research Directions and Knowledge Gaps

The final research question concerns what kind of knowledge gaps or future research directions can be identified among the literature on Indigenous peoples, resilience, and climate change in Taiwan. In terms of knowledge gaps, a geographical skewness was identified, with southern Taiwan receiving substantially more attention than Taiwan's east coast. This can be partly attributed to Typhoon Morakot and the relatively low impact it had on Taiwan's east coast. Government-led projects to studying Typhoon Morakot and associated funding channels also contributed to this skewness. This ties in with the types of climate hazards studied currently. Many scholars focus on climate shocks, as can be seen in the large overlap of Morakot's destruction path and the study sites of the selected

articles (Figure 6a; Appendix D). More studies must be conducted on climate stressors affecting Indigenous farmers and smallholders, how Indigenous peoples cope with climate variability, and how Indigenous peoples experience, perceive, and cope with global climate change. These stressor studies would help improve the east coast's Indigenous population's representation in future studies. This relates to a broader issue regarding the multiple impacts of climate change on Indigenous peoples in Taiwan. These could be direct (such as shocks), gradual (such as stressors), or indirect impacts, such as effects of climate change response by government agencies on Indigenous peoples. Highlighting the latter, Taiwan's quest for renewable or non-fossil fuel energy has led to new land grabs and disputes on traditional territories and rights of Indigenous peoples on Taiwan's east coast [76,77]. These multiple impacts, however, have not been comprehensively discussed by the articles in this bibliometric study.

In terms of the Indigenous peoples considered in the relevant studies, there are relatively few articles focusing on the Amis among other groups. The Amis are renowned for their TEK systems and local marine area management [20], and several questions remain regarding how climate change affects their traditional livelihoods and resilience. Future research could also adopt a longer timeframe, such as from 1990-2020, as studies prior to Typhoon Morakot could shed more light on communities that have received less attention, such as Indigenous communities on Taiwan's east coast.

We also discovered that relatively few studies have investigated how TEK systems can be integrated into climate change adaptation and disaster management. There is a valid concern that translating Indigenous languages into Chinese and then into English is difficult. Furthermore, some Indigenous elderly people are more fluent in Japanese than Chinese as a result of colonialization, and translation errors could also occur for this reason. Much information could be lost, and it, therefore, makes sense that these types of articles are published in Chinese in domestic outlets. Additionally, many (unpublished) masters theses—often indexed in Airiti Library—as well as grey literature on TEK have not been translated into academic articles yet. Nonetheless, the international community remains somewhat unexposed to the TEK systems of Taiwan's Indigenous peoples. This is probably another major reason why international literature has not covered Taiwan (e.g., [6]). In terms of methodological trends, many studies have adopted a qualitative approach, performing microlevel case studies. Although qualitative approaches provide rich data, they could be complemented with larger-scale quantitative surveys on the regional or national level. Comparative and longitudinal studies are also lacking. What are, for example, the long-term implications of climate-induced relocation? How does climate change create environmental mobility among Indigenous households? These questions require a more critical exploration of the relationship between Taiwan's Indigenous peoples, Indigenous resilience, and climate change. Perspectives in political ecology, for example, have been proven to be very valuable [78], but no study in our database adopted this approach. Future studies also need to adopt more multi-, cross-, and trans-disciplinary research methods. Studies on Indigenous peoples and climate change have been conducted from multiple academic disciplines, ranging from health sciences to DRR studies, and from ethnographic research to natural sciences. The next wave of Indigenous peoples' research should take advantage of these multiple disciplines by adopting more holistic approaches to the global climate change response of Indigenous peoples.

Indigenous resilience is also a dimension that deserves more attention in future studies. The concept of resilience consists of three dimensions: absorptive, adaptive, and transformative capacity, which correspond to coping, adaptation, and transformation respectively [79–82]. Most studies focused on coping or adaptation strategies [44,48,73], but no studies were found which analyzed the transformative capacity of Indigenous peoples to climate change. This consequently led to a somewhat narrow understanding of the concept of resilience. This is a research gap that should be addressed more comprehensively among studies on Indigenous peoples and climate change, both within the context of Taiwan and beyond.

It is crucial to mention why the lessons learned from Taiwan are important for other countries in the Asia-Pacific that are home to Indigenous peoples. Although Taiwan lacks international recognition, it is one of the few countries in Asia that officially recognizes its Indigenous peoples [19]. Taiwan is also a liberal democracy, and therefore the development path of Taiwan, being a newly developed country, can provide vital lessons for other countries in Asia that are home to Indigenous peoples, such as Vietnam, Cambodia, and Myanmar. It is hoped this study will prompt the international audience to engage more intensively with the literature on Indigenous peoples, resilience, and climate change in Taiwan. This study is the first of its kind for Taiwan's literature and is a humble beginning, but hopefully, the first step to give Taiwan's Indigenous peoples greater importance on the international stage.

Lastly, it is recommended that other studies conduct similar bibliometric analyses in countries home to the world's Indigenous peoples. The methodology employed in this study can be used to provide more insightful analyses embedded in national contexts to understand our current state of knowledge of how the world's Indigenous peoples cope and are resilient to global climate change.

## 4. Conclusions

This study analyzed 111 articles published in peer-reviewed domestic and international journals or books and concerning Indigenous peoples, resilience, and global climate change in Taiwan in the 10-year period after Typhoon Morakot. Most of the articles focused on disaster management, with a particular focus on post-disaster recovery, Indigenous cultures, ecological wisdom, and community development. Most studies were conducted within the context of and in relation to Typhoon Morakot, and more focus was given to climate shocks than climate stressors. There was also a somewhat narrow understanding of the concept of Indigenous resilience. Among the articles, we found a geographical skewness in favor of southern Taiwan, with Taiwan's eastern coast receiving relatively little attention. The geographical skewness could be partially attributed to the destruction path of Morakot, which overlapped considerably with the geographic locations of the included studies, and governmental research funding channels. The Amis, Taiwan's largest Indigenous group, were also largely overlooked. In terms of post-disaster recovery, most scholars argue for a more culturally sensitive approach that fits the needs and livelihoods of Indigenous peoples. Longitudinal studies and those focusing on all four phases of disaster management remain in their infancy. This also accounts for the small number of studies on TEK systems and climate change adaptation. Indigenous scholars have been very active in publishing their research, but most of their articles have been published in domestic outlets. Taiwan-based scholars should thus engage more with contemporary studies and global debates on the roles of Indigenous peoples in global climate change adaptation and mitigation. Taiwan is a case study and can provide the globe with an understanding of how Indigenous peoples can become more resilient to the negative effects of global climate change.

**Author Contributions:** All authors (M.M.B., Y.-Y.H., L.-S.H., H.-M.T. and t.'ev.) contributed to the study conception and design. Material preparation, data collection and analysis were performed by M.M.B. and Y.-Y.H. The first draft of the manuscript was written by M.M.B., Y.-Y.H., and L.-S.H. All authors commented on previous versions of the manuscript. All authors read and approved the final manuscript.

**Funding:** This study was supported by the Ministry of Science and Technology (MOST) of Taiwan under grant numbers MOST 109-2636-H-003-007 and MOST 108-2410-H-003-140.

**Acknowledgments:** We are very thankful to the comments of the three anonymous reviewers. Section 3.5 has been partly rewritten due to the helpful comments of Reviewer 2. We also thank Fikret Berkes for his comments on the first draft of this manuscript.

**Conflicts of Interest:** The authors declare no conflict of interest.

# Appendix A

**Table A1.** Indigenous population in Taiwan in 2020.

| Tribe | Indigenous Population | People in Indigenous Community |
|---|---|---|
| Atayal/Tayal | 92,306 | 47,531 |
| Saisiyat | 6743 | 2194 |
| Rukai | 13,498 | 8032 |
| Bunun | 59655 | 29,454 |
| Hla'alua | 418 | 1982 |
| Kanakanavu | 367 | 1018 |
| Tsou/Cou | 6709 | 3252 |
| Thao | 818 | 294 |
| Seediq | 10,485 | 8592 |
| Truku | 32,435 | 21,466 |
| Amis | 213,958 | 96,098 |
| Sakizaya | 992 | 1389 |
| Kavalan | 1503 | 373 |
| Puyuma | 14,573 | 7346 |
| Paiwan | 103,032 | 54,568 |
| Tao | 4692 | 4196 |
| Not-recognized | 10,902 | — |
| Total | 573,086 | 287,789 |

Source: Ministry of the Interior [72] and Council of Indigenous Peoples [18].

# Appendix B

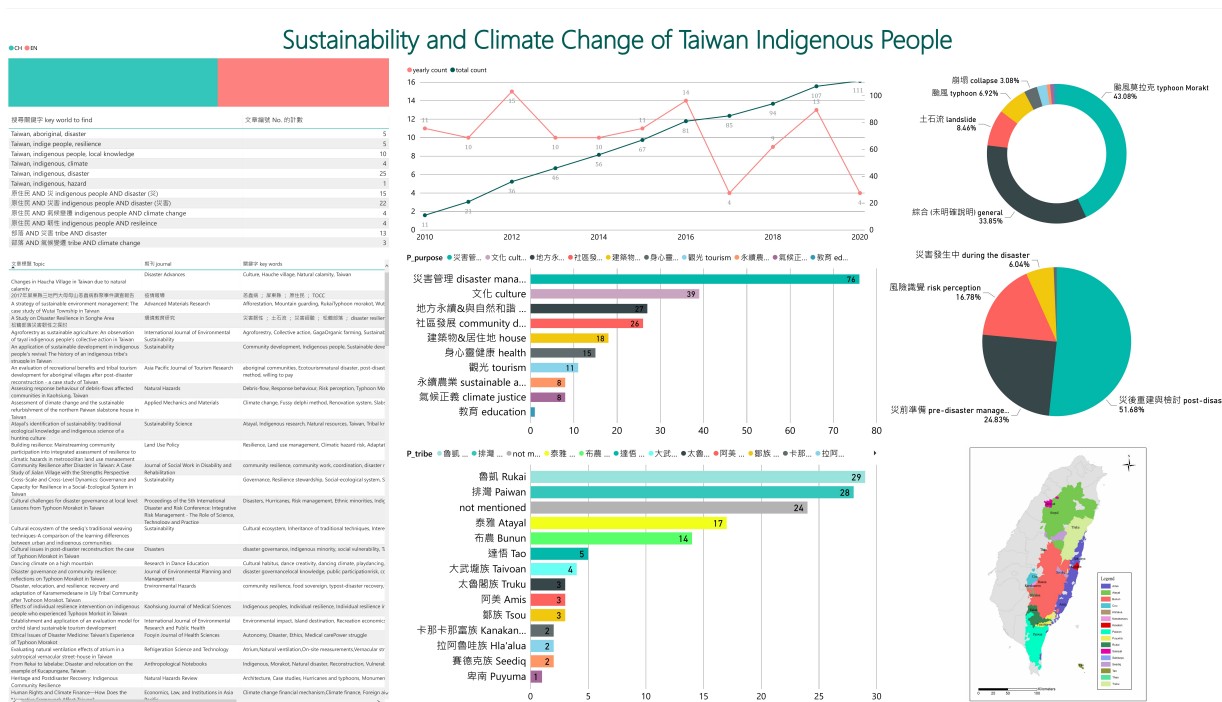

**Figure A1.** Screenshot of the Power BI analysis. Note: For the online version, see: https://bit.ly/3gyBW3b.

**Appendix C**

**Table A1.** List of all articles *.

| Code IN = International (Scopus) DM = Domestic (Airiti Library) | Full Bibliography: |
| --- | --- |
| DM01 | Chang A-S (2010) Dense Forest in the Past, but Few and Far between Nowadays the Dialogues between Aboriginals Traditional Mountain Experiences and Recent Mountain Development of Southern Region in Taiwan. Taiwan: A Radical Quarterly in Social Studies 78: 327–353. https://doi.org/10.29816/TARQSS.201006.0009 |
| DM02 | Chang W-C, Lin Y-S (2012) The Psychological Phenomenology of Natural Disaster Survivors: Cases from Typhoon Morakot. The Journal of Kaohsiung Behavioral Sciences 3:95–124. https://doi.org/10.29854/TJKBS.201205.0004 |
| DM03 | Chen C-F (2019) Regenerating "Home" between Auvinni Kadreseng's Writing of Storm Damage and the Construction of Social Resilience. Chung Wai Literary 48:169–194. https://doi.org/10.6637/CWLQ.201909_48(3).0005 |
| DM04 | Chen C-N, Yeh Y-L (2010) Recovery Evaluation of Hillslope Disaster in Mountain Tribes. Journal of Taiwan Agricultural Engineering 56:61–70. https://doi.org/10.29974/JTAE.201003.0005 |
| DM05 | Chen P-S, Li C-P, Tseng Y-L, Chuang H-H (2013) The study of disaster prevention social capital to construct innovative strategy in rural area—a case study of landslide disaster (in Chinese). Agricultural Extension Anthology 58: 89–122 |
| DM06 | Chen S-T, Hsu C-L, Kuo J-M, et al. (2011) Review on Disaster Preparedness and Response of Laiyi Township as Nanmadol Typhoon Period. Journal of Slopeland Hazard Prevention 10:37–48. https://doi.org/10.29995/JSHR.201112.0004 |
| DM07 | Chen T-C, Lin Y-S, Hsu W-Y (2013) An Exploration of the Differences between Aboriginals and Han in Their Coping Styles and Adaptation Strategies After Surviving Typhoon Morakot. Formosa Journal of Mental Health 26:249–278. https://doi.org/10.30074/FJMH.201306_26(2).0003 |
| DM08 | Chen W-L (2012) Reconstruction of Ka'aluwan Village after the Typhoon Morakot: A Case Study on Disaster of Anthropology. Chinese Journal of Applied Anthropology 1:157–173. https://doi.org/10.6290/CJAA.2012.0101.07 |
| DM09 | Chen Y-J (2011) A Community-Based Study on the Model of the Life Rehabilitation in the Aboriginal Disaster Areas. Cha Nan Annual Bulletin 37: 516–525. https://doi.org/10.29539/CNABH.201112.0019 |
| DM10 | Chen Y-L (2010) Subjectification, Movement, and Tribe Re-establishment in Indigenous Area of Southern Taiwan after Morakot Flood. Taiwan: A Radical Quarterly in Social Studies 78: 403–435. https://doi.org/10.29816/TARQSS.201006.0012 |
| DM11 | Chern J-C (2019a) Reconstruction after Typhoon Morakot: Achievements and Reflection on Its 10th Anniversary (I)—Prologue & Infrastructure Built for Disaster Prevention and Sustainability. Civil and Hydraulic Engineering 46:4–13. https://doi.org/10.6653/MoCICHE.201906_46(3).0001 |
| DM12 | Chern J-C (2019b) Reconstruction after Typhoon Morakot: Achievements and Reflection on Its 10th Anniversary (II)—Building a Colorful Sustainable Community. Civil and Hydraulic Engineering 46:14–30. https://doi.org/10.6653/MoCICHE.201906_46(3).0002 |
| DM13 | Chien H-F, Wang Y-C, Li H-C, et al. (2018) 2017年屏東縣三地門大母母山恙蟲病群聚事件調查報告[2017 Investigative Report for cluster infection of Scrub Typhus at Mt.Damumu in Sandimen Township, Pingtung County] (in Chinese). Taiwan Epidemiology Bulletin 34:115–118. https://doi.org/10.6524/EB.201804_34(7).0001 |

**Table A1.** *Cont.*

| Code IN = International (Scopus) DM = Domestic (Airiti Library) | Full Bibliography: |
|---|---|
| DM14 | Chien W-M (2011) 莫拉克風災對荖濃溪,楠梓仙溪流域原住民族群遷徙與文化變遷的影響與因應 [The impact and adaptation to the Indigenous groups' migration and culture change of Laonung river and Nanzixian river drainage basin by Typhoon Morakot] (in Chinese). Kaohsiung Historiography 1:6–27 |
| DM15 | Chiu FYL (2011) On Understanding Man-made Catastrophes and Re-generating Communal Capacities. Taiwan: A Radical Quarterly in Social Studies 85:317–352. https://doi.org/10.29816/TARQSS.201112.0008 |
| DM16 | Chuang P-F (2012) The Reconstruction and Healing Work after Typhoon Morakot. Journal of natural and human environment of Indigenous peoples. 3:55 -75 https://doi.org/10.29875/JNHEIP.201206.0003 |
| DM17 | Chung M-C, Tan C-H, Wang G-S, et al. (2010) Case Study of Ji-Lou Landslide Triggered by Typhoon Morakot. Journal of Chinese Soil and Water Conservation 41:333–342. https://doi.org/10.29417/JCSWC.201012_41(4).0005 |
| **DM18** | Dong X-H (2012) 莫拉克颱風嘉蘭村災後家園重建與社會文化的復振 [Post disaster reconstruction and cultural and social recovery of kaaluwan village after Typhoon Morakot] (in Chinese). Journal of natural and human environment of Indigenous peoples 3: 77–98. https://doi.org/10.29875/JNHEIP.201206.0004 |
| **DM19** | Du Chang M-C (2014) Indigenous Natural Resources Policy. Journal of The Taiwan Indigenous Studies Association 4:63–78 |
| **DM20** | Gadeljeman V, Taiban S (2019) Construction of Culture Space in Indigenous Community after Disaster: The Case Study of Three Permanent Housing Bases in Pingtun. Policy and Personnel Management 10:109–138. https://doi.org/10.29944/PPM.201906_10(1).0004 |
| DM21 | Hou Y-K, Liang B-K (2010) Tourism and Local Development of Indigenous Regions: A Case Study of Laiji Tribe of the Tsou People. Taiwan Journal of Indigenous Studies 3:105–148. https://doi.org/10.29910/TJIS.201003.0004 |
| DM22 | Hsia Y-J, Lin P-R (2011) Aborigine and Natural Resources Management: A Theoretical Framework for Co-management. Taiwan Journal of Indigenous Studies 4:39–66. https://doi.org/10.29910/TJIS.201103.0002 |
| DM23 | Hsieh W-C, Cheng S-F, Cheng C-W (2011) This Is Just a House, Not Our Home: The Immigration and Life-Shock's Experience of Taiwanese Indigene after Typhoon Morakot Through an Interpretive Interactionism Perspective. NTU Social Work Review 24: 135–166. https://doi.org/10.6171/ntuswr2011.24.04 |
| DM24 | Hsu C-L, Hsieh I-L, Huang H-C, Yan H-Y (2012a) Investigation of Potential Landslide Disaster on Bayao Tribe, Manzhou Township. Journal of Slopeland Hazard Prevention 11:1–13. https://doi.org/10.29995/JSHR.201212.0001 |
| DM25 | Hsu C-L, Liang J-Y, Chungku B-Y, et al. (2011) Causes of Chungya Tribe Slope Sliding in Luansan Village of Yanping Township at Taitung County. Journal of Slopeland Hazard Prevention 10:1–11. https://doi.org/10.29995/JSHR.201112.0001 |
| DM26 | Hsu C-L, Shu H-M, Chen S-T (2018) Discussion on the Adoption of Natural Ecological Engineering Method in Torrent Regulation. Journal of Slopeland Hazard Prevention 17:17–33 |
| DM27 | Hsu C-L, Son P-S (2011) The Investigation and Discussion on Vulnerability of Aboriginal Township in Pingtung County after Typhoon Morakot. Journal of Slopeland Hazard Prevention 10:1–9. https://doi.org/10.29995/JSHR.201106.0001 |
| DM28 | Hsu C-L, Yan H-Y, Hsieh I-L, Huang H-C (2012b) Investigation of Potential Landslide Disaster on Above Tribal of Kaoshih Village, Mudan Township. Journal of Slopeland Hazard Prevention 11:14–23. https://doi.org/10.29995/JSHR.201212.0002 |

**Table A1.** *Cont.*

| Code IN = International (Scopus) DM = Domestic (Airiti Library) | Full Bibliography: |
|---|---|
| DM29 | Hsu H-H, Lin P-SS (2018) Impact of Broadcast Equipment of Disaster Warning System and Disaster Experience on Residents' Adaptation Behavior: Case of Shuangchi Village, Taichung City. Journal of Disaster Management 7:113–134. https://doi.org/10.6149/JDM.2018.0701.05 |
| DM30 | Hung C-Y, Lin H-M (2010) A Study on the Residents' Hazard Perception and Adjustment Behavior for Wu-Fong and Jian-Shih Township in Hsinchu County. Journal of Engineering Environment 25:23–32. https://doi.org/10.6562/JEE.2010.25.2 |
| DM31 | Ku P-H, Shen C-W, Chen C-H, Tsai Y-L (2014) 原住民族部落之安全遷居地篩選原則與問題研析[Study of the selection principal and problem of safe relocation area for Indigenous tribes]. Sinotech Engineering 23–32 |
| **DM32** | Kuan D-W (2015) Indigenous Ecological Knowledge and Contemporary Disaster Management a Case Study on the Tayal Communities' Experience in the Watershed of Shih-Men Reservoir (in Chinese). Journal of Geographical Science 76:97–132. https://doi.org/10.6161/jgs.2015.76.04 |
| DM33 | Kuo T-H, Chen S-M (2017) Study of The Taiwan Paiwan tribe ecological environment adaptation: Tribal Chi Chia (tjuvecekadan) Case. Shih Chien Journal of Liberal Arts 25:75–100 |
| **DM34** | Lamuran S, Hsiao S-H, Tsai H-M (2015) Tao People's Response to Modern Environmental Governance and the Development of the Sustainable Environmental Governance. Journal of The Taiwan Indigenous Studies Association 5:1–44 |
| **DM35** | Lamuran S, vayayana tibusungu (2016) Tao Traditional Ecological Knowledge and Its Value for Sustainability. Journal of Geographical Research 65:143–167. https://doi.org/10.6234/JGR.2016.65.06 |
| DM36 | Lii D-T (2010) "Back-To-The-Land" the Reconstruction of a Disaster Society. Taiwan: A Radical Quarterly in Social Studies 78:273–326. https://doi.org/10.29816/TARQSS.201006.0008 |
| DM37 | Lin H-H, Chang C-Y, Huang YS (2019a) The Exploration of Community Identity through Tourism Development in Post-disaster Aboriginal Community. Journal of Island Tourism Research 12:1–27 |
| DM38 | Lin H-S, Yu M-C, Liu HK (2019b) Disaster Management and Development of Community Resilience: A Case Study of the DaHo Tribe in Pingtung County in Taiwan. Journal of Public Administration 57:1–38. https://doi.org/10.30409/JPA.201909_(57).0001 |
| **DM39** | Lin J-C (2012) A Discussion on the National Museum of Prehistory's Participating the Spiritual and Educational Reconstruction Project at Jialan Village After Disaster of Typhoon Morakot-Also on the Discussion of the Mutual Aid Mechanism of Tribes' Response to Disasters. Journal of natural and human environment of Indigenous peoples 3:35–54. https://doi.org/10.29875/JNHEIP.201206.0002 |
| DM40 | Lin J-J, Lin W-I (2014) A Study on the Service Networks of Typhoon Morakot Post-Disaster Reconstruction in the South Taiwan: A Governance Approach to Disaster. Thought and Words: Journal of the Humanities and Social Sciences 52:5–52 |
| DM41 | Lin W-Y, Li J-Y (2015) A Study on Community Industry Recovery Factors after Large-Scale Disaster-Using Xinkai Tribal Settlement at Xinfa Village, Liouguei District, Kaohsiung City after Typhoon Morakot as an Example. Journal of Disaster Management 4:1–29. https://doi.org/10.6149/JDM.2015.0401.01 |
| DM42 | Lin Y-S, Chang W-C (2017) Exploring Psychological Healing Experience of Paiwan Aboriginal Survivor after Typhoon Morakot: An Existential Phenomenology Approach. Monthly review of Philosophy and Culture 44:85–102 |

**Table A1.** *Cont.*

| Code IN = International (Scopus) DM = Domestic (Airiti Library) | Full Bibliography: |
|---|---|
| **DM43** | Mona A (2009) 氣候變遷,生態永續與原住民族社會文化發展:莫拉克風災的反思 [Reflection of Typhoon Morakot: climate change, ecological sustainability and socio-cultural development of Indigenous people] (in Chinese). Taiwan Indigenous Studies Review 6:27–54. https://doi.org/10.29763/TISR.200912.0002 |
| **DM44** | Shein PP, Huang Y-C, Chang M-C, Ruljigaljig T (2016) Part of a Whole: A Science Learning Ecosystem Perspective on Culturally Relevant Curriculum. Chinese Journal of Science Education 24:461–485. https://doi.org/10.6173/CJSE.2016.24S.02 |
| DM45 | Shieh J-C, Chen J-S, Lin W-I (2013) Skip to Permanence without Transition? Policy-making in Post-Morakot Reconstruction. Taiwan: A Radical Quarterly in Social Studies 93:49–86 |
| DM46 | Shieh J-C, Fu T-S, Chen J-S, Lin W-I (2012) A Road Far Away from the Aboriginal Hometown?-Rethinking the Post-disaster Relocation Policy of Typhoon Morakot. NTU Social Work Review 26:41–86. https://doi.org/10.6171/ntuswr2012.26.02 |
| DM47 | Sun C-T, Yen A-C (2012) An Investigation of Watershed Management and Land Ethics—A Case of Indigenous Communities on Upstream Shimen Reservoir Catchments Area. Journal of Geographical Science 66: 21–51. https://doi.org/10.6161/jgs.2012.66.02 |
| **DM48** | Taiban S (2012) Disaster, Relocation and Vulnerability: The Case Study of Kucapungane. Taiwan Journal of Anthropology 10:51–92. https://doi.org/10.7115/TJA.201206.0051 |
| **DM49** | Taiban S, Pei K, Lu T-C, et al. (2011) Conservation, Development and Relocation: Implementing the Forest Guard System Among the Western Rukai. Journal of The Taiwan Indigenous Studies Association 1:145–174. https://doi.org/10.6396/JTIS.201109.0145 |
| **DM50** | Tuhi Martukaw (2016) The Invisible Indigenous Peoples' Rights in Climate Change Negotiation. Journal of The Taiwan Indigenous Studies Association 6:123–136 |
| DM51 | Tung C-M, Lin W-Y, Tu T-Y, Tsai H-N (2015) Issues of Enforcing Post-Disaster Permanent Housing Policy from the Housing Development and Residents' Living Patterns Point of View-A Case Study of XinKai Tribe, Liouguei District, Kaohsiung City. Journal of Architecture 92:99–124. https://doi.org/10.3966/101632122015060092006 |
| **DM52** | Wang C-H, Chen J-C, Lee Z-Y, et al. (2019) A Study on the Evaluation Satoyama-model of Resilience Index Based on Taiwanese Aboriginal Tribe Culture and Ecological Sustainable Development—A Case Study of Nan-Feng Village. Journal of Tourism and Leisure Management 7:139–156. https://doi.org/10.6510/JTLM.201912_7(2).0012 |
| DM53 | Wang J-H (2013) Agricultural Adaptation to Climate Change among Highland Indigenous Tribes. Journal of the Agricultural Association of Taiwan 14:491–505. https://doi.org/10.6730/JAAT.201310_14(5).0005 |
| DM54 | Wang J-H (2016) Association among Climate Disaster, Area Capital and Agricultural Production of Indigenous Peoples. Journal of Taiwan Agricultural Research 65:286–295. https://doi.org/10.6156/JTAR/2016.06503.06 |
| DM55 | Wang J-H, Chen M-F (2015) Risk adaptation to climate disasters of Indigenous tribe: An action research approach. Journal of the Agricultural Association of Taiwan 16:197–211. https://doi.org/10.6730/JAAT.201509_16(3).0001 |
| DM56 | Wang Y-C, Lin S-W, Chen Z-E (2014) Application of Spatial Analysis to Explore the Evacuation Behavior of the Paratan Tribe in Hualien County. Bulletin of the Geographical Society of China 53:1–16. https://doi.org/10.29972/BGSC.201412_(53).0001 |
| DM57 | Wu C-L (2015) Erecting tata in front of Permanent House: Architectural Design and Social Reform for the Post-Disaster Reconstruction. Taiwanese Journal for Studies of Science Technology & Medicine 201504 (20): 9–73 |
| DM58 | Wu ZP, Hsiao HH (2014) A Study on the Effect of Refuge Decision during the Mudslide Disaster-A Case of Lai-Yi Village. Journal of Crisis Management 11:61–74. https://doi.org/10.6459/JCM.201403_11(1).0007 |

**Table A1.** *Cont.*

| Code IN = International (Scopus)<br>DM = Domestic (Airiti Library) | Full Bibliography: |
|:---:|:---|
| DM59 | Yeh K-H (2016) A Study on Disaster Resilience in Songhe Area. Journal of Environmental Education Research 12:73–104. https://doi.org/10.6555/JEER.12.2.073 |
| **DM60** | Zhang J-B (2012) Morakot Reconstruction to Reflect on the Trials and Rebirth: Lens. Journal of natural and human environment of Indigenous peoples 3:1–34. https://doi.org/10.29875/JNHEIP.201206.0001 |
| DM61 | Lin H-C, Lin H-T, Chou J-H (2010) An Exploration of Wind-Resisting Strategy on Traditional Yami Settlements. Journal of Architecture 73: 109–124. https://doi.org/10.6377/JA.201009.0109 |
| **IN01** | Ba Q-X, Lu D-J, Kuo WH-J, Lai P-H (2018) Traditional Farming and Sustainable Development of an Indigenous Community in the Mountain Area—A Case Study of Wutai Village in Taiwan. Sustainability (Switzerland) 10:3370. https://doi.org/10.3390/su10103370 |
| **IN02** | Berg KJ, Icyeh L, Lin Y-R, et al. (2016) Multiple-factor classification of a human-modified forest landscape in the Hsuehshan Mountain Range, Taiwan. Ambio 45:919–932. https://doi.org/10.1007/s13280-016-0794-5 |
| IN03 | Chen C-J (2013) The sustainable architectural design of post-disaster reconstruction of the aboriginal settlements in Taiwan. WIT Transactions on Ecology and the Environment 179 1:451–459. https://doi.org/10.2495/SC130381 |
| IN04 | Chen H-S (2019) Establishment and application of an evaluation model for orchid island sustainable tourism development. International Journal of Environmental Research and Public Health 16:. https://doi.org/10.3390/ijerph16050755 |
| IN05 | Chen M-H, Lin Y-J (2017) Integrating Co-management and the satoyama initiative for forest governance: Community-based ecotourism and conservation of Adiri and Labuwan. Taiwan Journal of Forest Science 32:299–316 |
| IN06 | Chen Y (2012) The Indigenous ecotourism and social development in Taroko National Park area and San-Chan tribe, Taiwan. GeoJournal 77:805–815. https://doi.org/10.1007/s10708-010-9373-7 |
| IN07 | Chen Y-L, Hsu W-Y, Lai C-S, et al. (2015) One-year follow up of PTSD and depression in elderly aboriginal people in Taiwan after Typhoon Morakot. Psychiatry and Clinical Neurosciences 69:12–21. https://doi.org/10.1111/pcn.12227 |
| IN08 | Chen Y-L, Lai C-S, Chen W-T, et al. (2011) Risk factors for PTSD after Typhoon Morakot among elderly people in Taiwanese aboriginal communities. International Psychogeriatrics 23:1686–1691. https://doi.org/10.1017/S1041610211000986 |
| IN09 | Cheng S-F, Cheng C-W, Hsieh W-C, et al. (2012) Effects of individual resilience intervention on Indigenous people who experienced Typhoon Morkot in Taiwan. Kaohsiung Journal of Medical Sciences 28:105–110. https://doi.org/10.1016/j.kjms.2011.10.015 |
| IN10 | Fan M-F (2015) Disaster governance and community resilience: reflections on Typhoon Morakot in Taiwan. Journal of Environmental Planning and Management 58:24–38. https://doi.org/10.1080/09640568.2013.839444 |
| IN11 | Fang W-T, Hu H-W, Lee C-S (2016) Atayal's identification of sustainability: traditional ecological knowledge and Indigenous science of a hunting culture. Sustainability Science 11:33–43. https://doi.org/10.1007/s11625-015-0313-9 |
| IN12 | Fu T-H, Lin W-I, Shieh J-C (2013) The Impact of Post-Disaster Relocation on Community Solidarity: The Case of Post-Disaster Reconstruction after Typhoon Morakot in Taiwan. International Journal of Structural and Construction Engineering 7:1870–1873. https://doi.org/doi.org/10.5281/zenodo.1072712 |

**Table A1.** *Cont.*

| Code IN = International (Scopus)<br>DM = Domestic (Airiti Library) | Full Bibliography: |
|---|---|
| IN13 | Gowlland G (2020) The materials of indigeneity: slate and cement in a Taiwanese Indigenous (Paiwan) mountain settlement. Journal of the Royal Anthropological Institute 26:126–145. https://doi.org/10.1111/1467-9655.13182 |
| IN14 | Hsu M (2016) Lost, found and troubled in translation: Reconsidering imagined Indigenous "communities" in post-disaster Taiwan settings. AlterNative 12:71–85. https://doi.org/10.20507/AlterNative.2016.12.1.6 |
| IN15 | Hsu M, Howitt R, Chi C-C (2014) The idea of "Country": Reframing post-disaster recovery in Indigenous Taiwan settings. Asia Pacific Viewpoint 55:370–380. https://doi.org/10.1111/apv.12058 |
| IN16 | Hsu M, Howitt R, Miller F (2015) Procedural vulnerability and institutional capacity deficits in post-disaster recovery and reconstruction: Insights from Wutai Rukai experiences of Typhoon Morakot. Human Organization 74:308–318. https://doi.org/10.17730/0018-7259-74.4.308 |
| IN17 | Huang CK, Lu CW (2013) Changes in Haucha Village in Taiwan due to natural calamity. Disaster Advances 6:22–37 |
| IN18 | Huang J-J, Shieh V, Huang M-Y, Lo HWA (2010) Ethical Issues of Disaster Medicine: Taiwan's Experience of Typhoon Morakot. Fooyin Journal of Health Sciences 2:94–97. https://doi.org/10.1016/S1877-8607(11)60005-2 |
| IN19 | Huang L-M (2013) Assessment of climate change and the sustainable refurbishment of the northern paiwan slabstone house in Taiwan. Applied Mechanics and Materials 311:527–532. https://doi.org/10.4028/www.scientific.net/AMM.311.527 |
| IN20 | Huang S-M (2018a) Understanding disaster (in)justice: Spatializing the production of vulnerabilities of Indigenous people in Taiwan: Environment and Planning E: Nature and Space. https://doi.org/10.1177/2514848618773748 |
| IN21 | Huang S-M (2018b) Heritage and Postdisaster Recovery: Indigenous Community Resilience. Natural Hazards Review 19:. https://doi.org/10.1061/(ASCE)NH.1527-6996.0000308 |
| IN22 | Hung H-C, Yang C-Y, Chien C-Y, Liu Y-C (2016) Building resilience: Mainstreaming community participation into integrated assessment of resilience to climatic hazards in metropolitan land use management. Land Use Policy 50:48–58. https://doi.org/10.1016/j.landusepol.2015.08.029 |
| IN23 | Hwang S-H, Huang H-M (2019) Cultural ecosystem of the seediq's traditional weaving techniques-A comparison of the learning differences between urban and Indigenous communities. Sustainability (Switzerland) 11: https://doi.org/10.3390/su11061519 |
| IN24 | Lee K-C, Yan S-Y (2019) Participatory planning and monitoring of protected landscapes: a case study of an Indigenous rice paddy cultural landscape in Taiwan. Paddy and Water Environment 17:539–548. https://doi.org/10.1007/s10333-019-00750-1 |
| IN25 | Lin H-L, Pan K-L (2018) VGI Cooperative Platform for Geo-information Crowdsourcing in Indigenous knowledge of community disaster resilience. In: 1st IEEE International Conference on Knowledge Innovation and Invention, ICKII 2018. Hotel Maison GladJeju Island, South Korea, pp 339–342 |
| IN26 | Lin J-J (2014) Cultural challenges for disaster governance at local level: Lessons from Typhoon Morakot in Taiwan. In: Proceedings of the 5th International Disaster and Risk Conference: Integrative Risk Management—The Role of Science, Technology and Practice, IDRC Davos 2014. Davos; Switzerland, pp 415–417 |
| IN27 | Lin J-J, Lin W-I (2016) Cultural issues in post-disaster reconstruction: the case of Typhoon Morakot in Taiwan. Disasters 40:668–692. https://doi.org/10.1111/disa.12172 |
| IN28 | Lin K-HE, Polsky C (2016) Indexing livelihood vulnerability to the effects of typhoons in Indigenous communities in Taiwan. The Geographical Journal 182:135–152. https://doi.org/10.1111/geoj.12141 |

**Table A1.** *Cont.*

| Code IN = International (Scopus) DM = Domestic (Airiti Library) | Full Bibliography: |
|---|---|
| IN29 | Lin P-SS, Chang K-M (2020) Metamorphosis from local knowledge to involuted disaster knowledge for disaster governance in a landslide-prone tribal community in Taiwan. International Journal of Disaster Risk Reduction 42: https://doi.org/10.1016/j.ijdrr.2019.101339 |
| IN30 | Lu Y (2017) The relationship between exhibitions about natural disasters and construction of ethnic identity in Taiwan. Sokendai Review of Cultural and Social Studies 13:239–255 |
| IN31 | Luo Y, Shaw R, Lin H, Joerin J (2014) Assessing response behaviour of debris-flows affected communities in Kaohsiung, Taiwan. Natural Hazards 74:1429–1448. https://doi.org/10.1007/s11069-014-1258-5 |
| **IN32** | Roder G, Ruljigaljig T, Lin C-W, Tarolli P (2016) Natural hazards knowledge and risk perception of Wujie Indigenous community in Taiwan. Natural Hazards 81:641–662. https://doi.org/10.1007/s11069-015-2100-4 |
| IN33 | Ru H-Y, Lo E-C (2015) The local moral world and agricultural activities of the committed organic farmer: A case study from an Atayal Community in Shilei, Jianshi Township, Xinzhu. Taiwan Journal of Anthropology 13:79–130 |
| IN34 | Shie Y-J (2020) Indigenous legacy for building resilience: A case study of Taiwanese mountain river ecotourism. Tourism Management Perspectives 33: https://doi.org/10.1016/j.tmp.2019.100612 |
| IN35 | Shih W-C (2019) Human Rights and Climate Finance—How Does the Normative Framework Affect Taiwan? In: Cohen JA, Alford WP, Lo C (eds) Taiwan and International Human Rights: A Story of Transformation. Springer, Singapore, pp 495–518 |
| IN36 | Su Y-M, Hsieh Y-C, Lin Y-C (2015) Evaluating natural ventilation effects of atrium in a subtropical vernacular street-house in Taiwan. In: Refrigeration Science and Technology. Nishi-kuYokohama, Japan, pp 4025–4032 |
| IN37 | Sun C-Y, Tai H-H, Yen A-C (2019) Use of planning training courses and activities to enhance the understanding of eco-community planning concepts in participatory planning workshop participants: A case study in taiwan. International Journal of Environmental Research and Public Health 16: https://doi.org/10.3390/ijerph16091666 |
| IN38 | Tai H-S (2015) Cross-Scale and Cross-Level Dynamics: Governance and Capacity for Resilience in a Social-Ecological System in Taiwan. Sustainability (Switzerland) 7:2045–2065. https://doi.org/10.3390/su7022045 |
| IN39 | Tai J (2018) Dancing climate on a high mountain. Research in Dance Education 19:294–305. https://doi.org/10.1080/14647893.2018.1523381 |
| **IN40** | Taiban S (2013) From Rekai to labelabe: Disaster and relocation on the example of Kucapungane, Taiwan. Anthropological Notebooks 19:59–76 |
| **IN41** | Taiban S, Gau H-S, Chung C-Y, et al., (2013) A strategy of sustainable environment management: The case study of Wutai Township in Taiwan. Advanced Materials Research 684:207–211. https://doi.org/10.4028/www.scientific.net/AMR.684.207 |
| **IN42** | Taiban S, Lin H-N, Ko C-C (2020) Disaster, relocation, and resilience: recovery and adaptation of Karamemedesane in Lily Tribal Community after Typhoon Morakot, Taiwan. Environmental Hazards 19:209–222. https://doi.org/10.1080/17477891.2019.1708234 |
| IN43 | Tang C-P, Tang S-Y (2010) Institutional adaptation and community-based conservation of natural resources: The cases of the Tao and Atayal in Taiwan. Human Ecology 38:101–111. https://doi.org/10.1007/s10745-009-9292-8 |
| IN44 | Wang L-R, Chen S, Chen J (2013) Community Resilience after Disaster in Taiwan: A Case Study of Jialan Village with the Strengths Perspective. Journal of Social Work in Disability and Rehabilitation 12:84–101. https://doi.org/10.1080/1536710X.2013.784551 |

**Table A1.** *Cont.*

| Code IN = International (Scopus)<br>DM = Domestic (Airiti Library) | Full Bibliography: |
|---|---|
| IN45 | Wu H-C (2014) Protectors of Indigenous Adolescents' Post-disaster Adaptation in Taiwan. Clinical Social Work Journal 42:357–365. https://doi.org/10.1007/s10615-013-0448-z |
| IN46 | Wu HC (2013) The predictors of new life adaptation for adolescents after Typhoon Morakot. In: Psychology of Trauma. pp 151–166 |
| IN47 | Wu S-T, Chiu C-H, Chen Y-S (2019) An evaluation of recreational benefits and tribal tourism development for aboriginal villages after post-disaster reconstruction—a case study of Taiwan. Asia Pacific Journal of Tourism Research 24:136–149. https://doi.org/10.1080/10941665.2018.1556710 |
| IN48 | Yen A-C, Chen Y-A (2014) Sustainable agriculture and Indigenous community development: Some experiences in Taiwan. International Journal of Sustainability in Economic, Social, and Cultural Context 9:85–105. https://doi.org/10.18848/2325-1115/cgp/v09i03/55240 |
| IN49 | Yen A-C, Chen Y-A (2016) Agroforestry as sustainable agriculture: An observation of tayal Indigenous people's collective action in Taiwan. International Journal of Environmental Sustainability 13: |
| IN50 | Yu C-Y (2018) An application of sustainable development in Indigenous people's revival: The history of an Indigenous tribe's struggle in Taiwan. Sustainability (Switzerland) 10:9. https://doi.org/10.3390/su10093259 |

\* Note: we present this bibliography in English according to the English titles provided by the authors themselves. In case there were only Chinese titles available, we translated them ourselves but we also provided the original titles in Chinese. It is also important to mention that not all articles had DOI numbers, and therefore we only included them if they were available. Codes in **bold** have been (co-)written by Indigenous authors.

**Appendix D**

Geographical distribution of all articles and damage of Typhoon Morakot in 2009 and damage of extreme climate events on a national level from 2006 till 2020 in Taiwan.

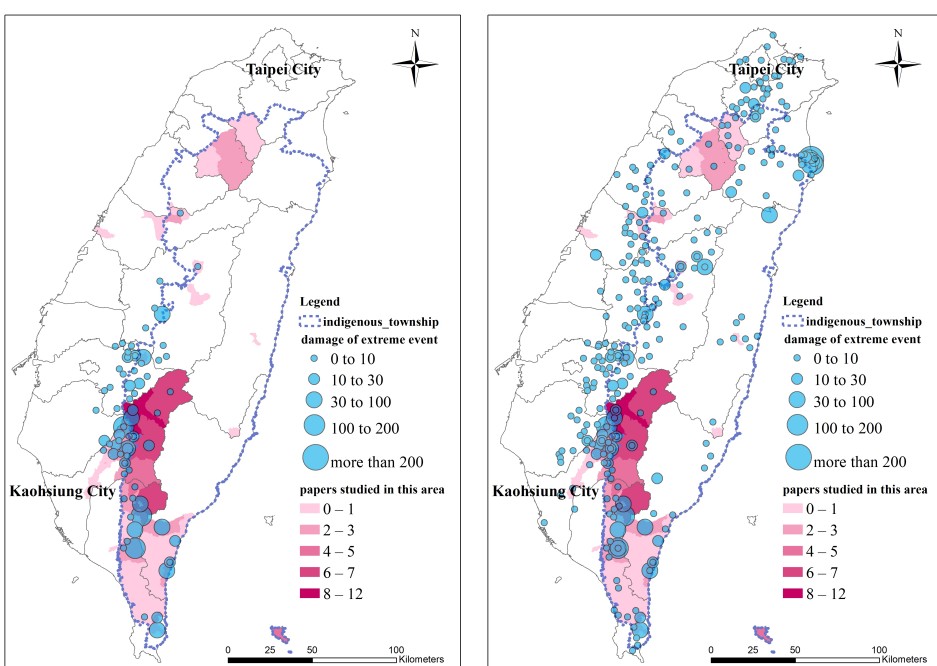

**Figure A2.** Cumulative housing damage from typhoon Morakot (**a**) and of all extreme climate events (**b**) in Taiwan (data from: [24]).

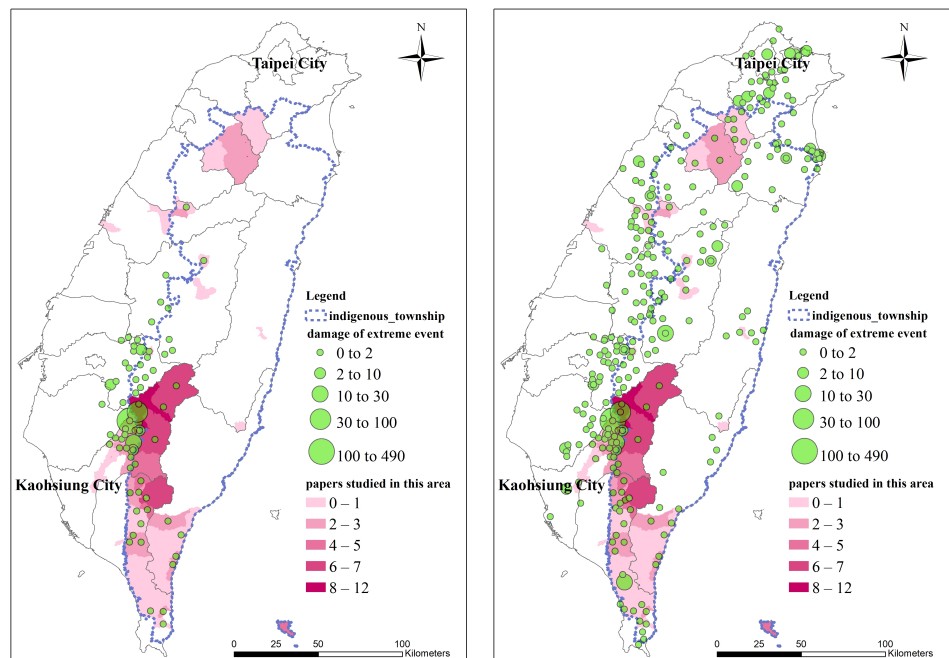

**Figure A3.** Cumulative score of deceased and injured victims of typhoon Morakot (**a**) and of all extreme climate events (**b**) in Taiwan (data from: [24]).

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
