# Peer review of "Global Climate Change and Indigenous Peoples in Taiwan: A Critical Bibliometric Analysis and Review"

_sustainability, doi:10.3390/su13010029_

Round 1
Reviewer 1 Report
Overall really interesting paper on an interesting topic, very well written. A few improvements could be made to the figures and tables:
Figure 1: very difficult to read, even after zooming in. Consider a different disposition (e.g. bigger pictures, one above the other rather than side by side).
Table 2: I would suggest to again specify IN and DM in the caption (and try to have caption and table on the same page).
Figure 4: the words listed in the text above are not the same as the most evident words from the word clouds. For 4a, it seems to be: disaster, indigenous, community, Morakot, Taiwan, Sustainable (and many others, but I don’t even see “resilience” and “sustainability”). For 4b: Disaster, Morakot, Typhoon, indigenous, community. How come is there such a mismatch between the text and the figures?
Figure 5: same problem as with the other map, especially the legend is not readable at all.
Figure 6: same as above. It is a pity, because a lot of the information (especially about the damage of the extreme event) is lost, completely undistinguishable on the map.
Figure 7: comes a bit out of the blue, I would consider moving it below the paragraph referring to it.
Author Response
We are very thankful to Reviewer 1 for reviewing our manuscript. We will now highlight the comments point-wise. Changes in the text are marked in yellow.
“Overall really interesting paper on an interesting topic, very well written. A few improvements could be made to the figures and tables:”
Thank you very much. We really appreciate it.
“Figure 1: very difficult to read, even after zooming in. Consider a different disposition (e.g. bigger pictures, one above the other rather than side by side).”
We agree, we modified it and we hope that during the proofing stage of the manuscript, the figures can be shown in a better and visually appealing way.
“Table 2: I would suggest to again specify IN and DM in the caption (and try to have caption and table on the same page).”
We modified it.
“Figure 4: the words listed in the text above are not the same as the most evident words from the word clouds. For 4a, it seems to be: disaster, indigenous, community, Morakot, Taiwan, Sustainable (and many others, but I don’t even see “resilience” and “sustainability”). For 4b: Disaster, Morakot, Typhoon, indigenous, community. How come is there such a mismatch between the text and the figures?”
We changed the text to fit it better with the figure (line 219-223).
“Figure 5: same problem as with the other map, especially the legend is not readable at all.”
We modified figure 5 now.
“Figure 6: same as above. It is a pity, because a lot of the information (especially about the damage of the extreme event) is lost, completely undistinguishable on the map.”
We modified figure 6 now. We tried to make it more distinguishable, but actually the main purpose of the figure is to show the overlap in studies and the destruction path of Morakot/climate disasters.
“Figure 7: comes a bit out of the blue, I would consider moving it below the paragraph referring to it.”
We now moved it.
Thank you very much for your sharp observations. We hope that the revisions made are done in a satisfactory manner.
Reviewer 2 Report
Please see the attached reviewer's comment.

Author Response
We are very thankful to Reviewer 2 for reviewing our manuscript. We will now highlight the comments point-wise. Changes in the text are marked in yellow.
“This paper analyzes the research on the indigenous peoples of Taiwan in the past ten years, especially concerning the issues of global climate change and resilience. The analysis of the paper is detailed and all the academic literature is collected. The methodology of the analysis has been clearly explained, and the results of the analysis are clearly presented. I agree with the analysis results already listed in this paper. On the other hand, for the implication and recommendation for analysis results, I have a few suggestions.
Regarding the gaps and areas that need to be strengthened in the study of Taiwan’s indigenous peoples revealed by the analysis results, I suggest that we can further rethink the following three perspectives.”
Thank you very much for your comments and suggestions. We really appreciate it.
“First, the impact of global climate change is comprehensive and not limited to the impact caused by typhoons. While most studies in Taiwan have focused on the follow-up effects of typhoon Morakot, global climate change actually causes more different and slow changes. And these shocks are not limited to agriculture only, but a comprehensive shock to the indigenous peoples. For example, in eastern Taiwan, the development of renewable energy in response to climate change has caused a new wave of invasions and disputes on the traditional territories and rights of the indigenous people. This type of new and long-term impact is actually as important as the impact of natural disasters, but so far, it has hardly been discussed by the Indigenous studies in Taiwan. In other words, if this article hopes to explore the gaps in the current research on indigenous peoples in more depth, then it is necessary to consider the multiple impacts of climate change.”
We fully agree. We now added the following sections to address this: “It is important to […] by global climate change.” (Line 124-130) and “This relates to a broader issue […] this bibliometric study.” (Line 409-415). Thank you for this suggestion!
“Second, the concept of resilience actually involves three important sub-concepts: coping, adaptation, and transformation. While most studies on indigenous resilience mainly focus on coping, few scholars studied the aspect of adaptation. There is almost no research on transformation. When the issue of transformation is not discussed, this will narrow people's thinking about resilience. This will in turn narrow the narrative, agency and options of indigenous peoples in face of climate change. Indigenous peoples are not only victims of typhoons. In fact, as the guardians of Taiwan’s main natural system, indigenous peoples are the main contributors to the fight against climate change. This transformation in discourse and practical action in indigenous society is expected to be, and should be the focus of the next wave of indigenous peoples research. This also reveals an important issue: in the face of the impact of climate change, whether and how Taiwan’s indigenous societies undergo, and actively pursue transformation (instead of only taking actions of coping and adaptation)? However, it is a pity that the study of indigenous peoples in Taiwan has not yet systematically discussed this aspect. This is a major research gap. And this gap comes from people’s narrow understanding about the concept of resilience.”
Another excellent suggestion! Thank you! We added this discussion now: “Most studies also adopted a somewhat narrow focus on Indigenous resilience.” (Line 27-28), “Studies focusing […] remain understudied.” (Line 226-228), “Indigenous resilience is also […] Taiwan and beyond.” (Line 447-454), and “There was also a somewhat narrow understanding of the concept of Indigenous resilience.” (Line 475-476).
“Last but not least, in order to study climate change and resilience issues, multi-, cross-, and even trans-disciplinary research is necessary. This raises an important question: in Taiwan’s current indigenous research literature, how many studies have the characteristics of multi-, cross-, and trans-disciplinary research?”
Third excellent suggestion, thank you! Please see: “Future studies […] of Indigenous peoples” (Line 441-446).
“I sincerely hope that the above mentioned three comments can assist the authors in identifying the gaps in the Taiwan's indigenous studies and the direction of future efforts of academic community.”
Thank you, as a sign of gratitude to you and the two other reviewers, we added: “Acknowledgments: We are very thankful to the comments of the three anonymous reviewers. Section 3.5 has been partly rewritten due to the helpful comments of Reviewer 2.” (Line 497-498).
Thank you very much for your sharp observations. We hope that the revisions made are done in a satisfactory manner.
Reviewer 3 Report
This manuscript provides a comprehensive review of the climate change impact on the lifestyles and cultures of Indigenous peoples in Taiwan. The authors focused on Typhoon Morakot, which caused severe damage to Taiwan's communities, and surveyed relevant peer-reviewed papers. A good point of this study is that it covers Chinese-language articles of the domestic journals in addition to English-language articles of the international journals. Studies on local communities are often published in the researchers' native language, but most foreign researchers cannot access the results due to the language barrier. This problem frequently occurs in Asian countries. The authors compared the articles published in the domestic and international journals and summarized the differences in the research keywords, target districts, disaster types, disaster management stages, and cooperation with Indigenous authors. This manuscript is well-organized based on a sufficient number of previous studies. The authors' description is straightforward, and the graphic images clearly show the results of the bibliometric analysis. The authors seem to pay adequate attention to ethnic diversity in Taiwan. For these reasons, the Reviewer concluded that the current version of the manuscript is suitable for publication in Sustainability.
Author Response
We are very thankful to Reviewer 3 for reviewing our manuscript. We will now highlight the comments point-wise. Changes in the text are marked in yellow.
“This manuscript provides a comprehensive review of the climate change impact on the lifestyles and cultures of Indigenous peoples in Taiwan. The authors focused on Typhoon Morakot, which caused severe damage to Taiwan's communities, and surveyed relevant peer-reviewed papers. A good point of this study is that it covers Chinese-language articles of the domestic journals in addition to English-language articles of the international journals. Studies on local communities are often published in the researchers' native language, but most foreign researchers cannot access the results due to the language barrier. This problem frequently occurs in Asian countries. The authors compared the articles published in the domestic and international journals and summarized the differences in the research keywords, target districts, disaster types, disaster management stages, and cooperation with Indigenous authors. This manuscript is well-organized based on a sufficient number of previous studies. The authors' description is straightforward, and the graphic images clearly show the results of the bibliometric analysis. The authors seem to pay adequate attention to ethnic diversity in Taiwan. For these reasons, the Reviewer concluded that the current version of the manuscript is suitable for publication in Sustainability.”
Thank you very much for the really great summary on our study as well as recognizing the need for this study. We highly appreciate it! Thank you!